# Model Optimization and In Silico Analysis of Potential Dipeptidyl Peptidase IV Antagonists from GC-MS Identified Compounds in *Nauclea latifolia* Leaf Extracts

**DOI:** 10.3390/ijms20235913

**Published:** 2019-11-25

**Authors:** Franklyn Nonso Iheagwam, Olubanke Olujoke Ogunlana, Shalom Nwodo Chinedu

**Affiliations:** 1Department of Biochemistry, Covenant University, PMB 1023, Ota 112212, Ogun State, Nigeria; banke.ogunlana@covenantuniversity.edu.ng (O.O.O.); shalom.chinedu@covenantuniversity.edu.ng (S.N.C.); 2Covenant University Public Health and Wellness Research Cluster (CUPHWERC), Covenant University, PMB 1023, Ota 112212, Ogun State, Nigeria

**Keywords:** gas chromatography-mass spectroscopy, *Nauclea latifolia*, dipeptidyl peptidase IV, in silico, homology modeling, molecular docking, ADMET

## Abstract

Dipeptidyl peptidase IV (DPP-IV) is a pharmacotherapeutic target in type 2 diabetes. Inhibitors of this enzyme constitute a new class of drugs used in the treatment and management of type 2 diabetes. In this study, phytocompounds in *Nauclea latifolia* (NL) leaf extracts, identified using gas chromatography–mass spectroscopy (GC-MS), were tested for potential antagonists of DPP-IV via in silico techniques. Phytocompounds present in *N. latifolia* aqueous (NLA) and ethanol (NLE) leaf extracts were identified using GC–MS. DPP-IV model optimization and molecular docking of the identified compounds/standard inhibitors in the binding pocket was simulated. Drug-likeness, pharmacokinetic and pharmacodynamic properties of promising docked leads were also predicted. Results showed the presence of 50 phytocompounds in NL extracts of which only 2-*O*-p-methylphenyl-1-thio-β-d-glucoside, 3-tosylsedoheptulose, 4-benzyloxy-6-hydroxymethyl-tetrahydropyran-2,3,5-triol and vitamin E exhibited comparable or better binding iGEMDOCK and AutoDock Vina scores than the clinically prescribed standards. These four compounds exhibited promising drug-likeness as well as absorption, distribution, metabolism, excretion and toxicity (ADMET) properties suggesting their candidature as novel leads for developing DPP-IV inhibitors.

## 1. Introduction

Type 2 diabetes mellitus (T2DM) is a chronic metabolic disease whereby various physiologic and metabolic pathways that occur simultaneously to ensure glycemic homeostasis are affected, truncated and worsen over time due to insulin resistance and beta-cell function decline [1]. In patients suffering this disease, incretin hormones levels are reduced significantly affecting various biological and physiological activities they carry out such as insulinotropic effects [2]. Glucagon-like peptide-1 (GLP1) and glucose-dependent insulinotropic peptide (GIP) are two major incretin hormones that mediate glycemic control via satiety promotion, glucagon secretion suppression and gastric emptying rate reduction [3,4,5,6]. They are responsible for about 50% to 65% insulin secreted during ingestion but this effect does not last due to their extremely short half-lives following secretion [7,8]. This metabolic characteristic is as a result of rapid degradation by dipeptidyl peptidase IV (DPP-IV), resulting in loss of their insulinotropic activities. Dipeptidyl peptidase IV (DPP-IV; EC 3.4.14.5) was first identified by Hopsu-Havu and Glenner in the year 1966 as a dipeptide naphthylamidase with other nomenclatures such as lymphocyte cell surface marker CD26 and adenosine deaminase (ADA)-binding protein [9,10]. DPP-IV is a 110 kDa glycoprotein, serine protease, membrane-associated ectoenzyme, made up of 766 amino acids, whose mechanism of action is by cleaving the N-terminal dipeptide end of its substrates that contain either proline or an alanine residue at the penultimate position [11,12,13]. Its wide distribution in various tissues and sites (bone marrow, small intestine, kidney, liver, blood vessels, body fluids, epithelial and endothelial cells) in the body, as well as reported effects on energy metabolism and insulin action has made DPP-IV become an interesting pharmacotherapeutic target in diabetes mellitus (DM) treatment and management [13,14,15]. In order to maintain optimal glycemic control in DM patients, maintain hormonal insulinotropic activities and overcome the degradation of these hormones, DPP-IV inhibitors (DPP-IVi) have been used as monotherapy or polytherapy with other antidiabetic drugs [16,17]. These drugs have been clinically formulated based on the structure related function of the inhibitors and improved knowledge of the enzyme. These drugs are classified into reversible and covalently modifying product analogue inhibitors based on their mechanism of action or pyrrolidines, cyanopyrrolidines (peptidic mimetics) and heterocyclics (non-peptidic mimetics) based on their structure [18,19]. Sitagliptin, vildagliptin, saxagliptin, alogliptin and linagliptin developed by Merck, Novartis, Bristol-Myers Squibb, Takeda and Boehringer Ingelheim respectively are the common DPP-IVi approved and sold in the market worldwide under various names [11,20].

Medicinal plants have been used by various populations worldwide for the management and treatment of diabetes mellitus (DM). However, the increased interest in traditional and complementary medicine has led to the identification, validation and utilization of bioproducts derived from plants for the treatment and management of diseases such as DM [13]. *Nauclea latifolia* (NL) is one of such medicinal plants with reported and validated antidiabetic activity, which has been attributed to the high alkaloid content and other biological activities [21,22,23,24]. Antimalarial [25], antihelmintic, antimicrobial [26], antiviral [27], antiulcer [28], analgesic [29], antioxidant and hepatoprotective [30] activities are some of the reported biological and pharmacological activities of different *N. latifolia* parts. On phytochemistry, β-sitosterol and its derivative, quinovic acid, β-carbolines, tramadol, scopoletin, p-coumaric acid, resveratrol, naucleol and other phyto bioactives have been isolated, characterized and confirmed to possess various pharmacological activity in various disease conditions [31]. Despite these broad reports, the limited number of approved drugs on a yearly basis is evidence of the challenging task behind the identification of novel lead compounds [13]. Hence, identification of effective and novel DPP-IVi from natural products for the management of type 2 diabetes is a dynamic area of research due to the general consideration of little to no toxicity, lower side effects and cost compared to synthetic medications [13]. This study implemented in silico techniques to discover potential DPP-IV antagonists from GC-MS identified compounds in *Nauclea latifolia* leaf extracts.

## 2. Results

### 2.1. Gas Chromatography-Mass Spectroscopy (GC-MS) Results

The gas chromatogram of *N. latifolia* ethanol (NLE) and aqueous (NLA) leaf extracts showed the presence of 47 and 21 peaks respectively (Figure 1 and Figure 2).

From the peaks, 41 phytocompounds ranging from 2-furanmethanediol, dipropionate (5.702) to 17-octadecynoic acid (20.721) were identified for NLE based on their mass spectra and retention time (Table 1). For NLA, 19 phytocompounds were identified ranging from 2,3-butanediol (5.805) to 9,12-octadecadienoic acid (Z,Z; 19.217) based on their mass spectra and retention time (Table 2). 2-furanmethanol and 2-oxopentanedioic acid were the least abundant phytoconstituents with 0.09% while octadecanoic acid, ethyl ester was the most abundant with 18% in NLE while phytol (0.07%) and [1,1’-bicyclopropyl]-2-octanoic acid, 2’-hexyl- and methyl ester (20.04%) were observed to be the least and most abundant phytoconstituents in NLA respectively (Table 1 and Table 2). Carboxylic acids, alcohols, alkaloids, carbohydrates, fatty acid, terpenes/terpenoids and phenolics made up 2%, 5%, 5%, 10%, 17%, 27% and 34% respectively of the identified compounds in NLE as depicted in Table 1 while in Table 2, alcohols, fatty acid, phenolics and terpenes/terpenoids made up 11%, 26%, 42% and 21% of the identified compounds in NLA. However, 10 phytocompounds such as phytol, n-hexadecanoic acid, 2-methoxy-4-vinylphenol and others were present in both extracts (Table 1 and Table 2). The mass spectroscopy (MS) spectra of both NLE and NLA GC chromatogram further corroborate the results (Appendix A).

### 2.2. Protein Sequence Retrieval and Model Optimization Results

The obtained DPP-IV sequence from NCBI with reference sequence identification number NP_001926.2 was made up of 766 amino acid residues. Querying of the sequence generated other proteins with a similar sequence. SWISS-MODEL also corroborated the result. Eleven *Homo sapiens* DPP-IV templates (1wcy, 3qbj, 2qt9, 2bgr, 5lls, 2gbg, 2jid, 1orv, 3f8s, 5vta and 4ffv) were selected but 1wcy was further chosen as the homology modeling template due to the sequence identity and similarity, global model quality estimate (GMQE), template resolution, quaternary structure quality estimate (QSQE), oligomeric state, local quality estimate and experimental comparison plot superiority over other templates (Table 3). Chain A of the modeled DPP-IV protein was selected despite structural similarities between the chains A and B. The modeled protein had a GMQE and QSQE score of 0.99 and 1 respectively. The protein was a homo-dimer with a 2.2 Å resolution and 0.62 sequence similarity (Table 3). The model also had a Z-score that was less than 1 (Z-score < 1) when compared with other pdb structures and a QMEAN of −0.56. The local quality estimate ranged from 0.7–0.9 with a few outliers lower than 0.6 (Figure 3).

### 2.3. Energy Minimization, Physicochemical Analysis and Model Evaluation Results

3D refine generated five energy-minimized models from the modeled DPP-IV of which model 5 was selected for further analysis based on the 3D refine score, RWplus and MolProbity (Table 4). When this energy-minimized model was superimposed with the modeled DPP-IV and 1wcy the root mean square deviation (RMSD) score was 0.262 and 0.267 Å respectively (Figure 4).

From the physicochemical analysis of the selected model, the protein has an atomic composition of 3832 carbons, 5721 hydrogens, 983 nitrogens, 1132 oxygens and 26 sulphurs (Figure 5). The observed atomic compositions was as a result of the amino acid compositions where serine (8.6%) followed by tyrosine (7.7%) and leucine (7.5%) were the highest amino acid residues with 63, 56 and 55 residues respectively while cysteine (1.6%) followed by methionine (1.9%) and histidine (2.6%) were the lowest amino acid residues with 12, 14 and 19 residues respectively (Figure 6). This protein was acidic with a molecular weight of 84,506.05 and an isoelectric point of 5.61. Additionally, an extinction coefficient of 194,190 M^−1^ cm^−1^ with an estimated half life of 1.1 h, 3 min and over 10 h in mammals, yeast and *Escherichia coli* respectively. 44.19 and 76.87 were recorded as the instability and aliphatic index respectively with a −0.407 grand average of hydropathicity.

The Ramachandran plot of the minimized modeled DPP-IV showed 88.8%, 10.6%, 0.3% and 0.3% of the amino acid residues were in the most favored, additional allowed, generously allowed and disallowed region respectively as compared to the modeled protein (89.6%, 10.1%, 0.2% and 0.2% respectively) and template (1wcy) residues (86.7%, 12.9%, 0.3% and 0.2% respectively) found in the most favored, additional allowed, generously allowed and disallowed region. The torsion angles (−0.02), covalent geometry (−1.07) and overall average (−0.52) of the energy minimized model were reduced compared to that of modeled DPP-IV (−0.20, 0.11 and −0.07 respectively) and the template 1wcy (0.13, 0.54 and 0.30 respectively; Table 5).

The ERRAT quality factor of the minimized modeled DPP-IV was 92.93% as shown in Figure 7 while from Figure 8, 95.34% was the recorded Verify 3D score. 

### 2.4. Pocket Identification and Molecular Docking Simulation Results

The potential binding pocket of the minimized modeled DPP-IV 3D structure was identified based on the pocket volume and druggability of the pocket in Figure 9 as simulated using DoGSiteScorer and PockDrug prior docking simulations.

### 2.5. Molecular Docking Simulation Results

The docking results for iGEMDOCK ranged from −50 to −90.22 kcal/mol for the GC–MS identified phytocompounds while −82.55 and −76.74 kcal/mol were recorded respectively for alogliptin and saxagliptin, which are clinically prescribed inhibitors of DPP-IV as summarized in Table 6. For AutoDock Vina, nine conformations were obtained for each phytocompounds and standard with the binding energy of the most stable conformation recorded in Table 6. The binding energy of the phytocompounds ranged from −4.3 to −7.8 kcal/mol while alogliptin and saxagliptin both had a score of −6.7 kcal/mol. It was further observed from Table 6 that 2-*O*-p-methylphenyl-1-thio-β-d-glucoside, 3-tosylsedoheptulose, 4-benzyloxy-6-hydroxymethyl-tetrahydropyran-2,3,5-triol and vitamin E had better iGEMDOCK (−76.67, −79.45, −90.22 and −77.10 kcal/mol respectively) and AutoDock Vina (−6.9, −7, −6.8 and −6.7 kcal/mol respectively) docking scores compared to either standard and were processed further as hits. 3-*O*-methyl-d-glucose had a relatively comparable iGEMDOCK score of −79.56 kcal/mol with the standard saxagliptin but a low AutoDock Vina score was recorded. The inverse was the case for androstan-17-one,16,16-dimethyl-(5-alpha)- and γ-sitosterol, whereby their AutoDock Vina score (−7.3 and −7.8 kcal/mol respectively) was way higher than both standards, but lower scores were observed when docking was carried out using iGEMDOCK.

The various binding poses of 2-*O*-p-methylphenyl-1-thio-β-d-glucoside, 3-tosylsedoheptulose, 4-benzyloxy-6-hydroxymethyl-tetrahydropyran-2,3,5-triol, vitamin E, alogliptin and saxagliptin as simulated by iGEMDOCK (Figure 10) and AutoDock Vina (Figure 11) were stabilized by conventional hydrogen, carbon-hydrogen, Van der Waals and pi–pi interactions between the ligands and various amino acid residues in the binding site. Lys122, Gln123, Trp124, Ser 630, Asp709, Asp739 and Ala743 were common amino acid residues involved in stabilizing these binding poses using iGEMDOCK (Table 7) while Glu205, Glu206, Tyr666 and Phe357 were synonymous with the stability of the poses simulated by AutoDock Vina (Table 8).

### 2.6. Druglikeness, Pharmacokinetic and Toxicity Prediction

From the physicochemical properties of the docked ligands that were identified as potential leads in Table 9, no potential leads violated Lipinski drug-likeness RO5 except vitamin E with a high octanol-water partition coefficient (LogP) of 8.84. Other parameters such as molecular weight, LogP, hydrogen bond acceptor and donor were within the specified limit of Lipinski drug-likeness.

From the predicted absorption properties tabulated in Table 10, the selected leads showed no human intestinal absorption and Caco-2 permeability except vitamin E. Nonetheless, they possessed blood–brain barrier permeability with 3-tosylsedoheptulose being the only exception. The compounds were also non-inhibitors of p-glycoprotein, renal organic cation transport and non-substrates of p-glycoprotein. The compounds were also not orally bioavailable at both 20 and 30% except 2-*O*-p-methylphenyl-1-thio-β-d-glucoside, which was orally bioavailable at 20%. For the distribution pattern of the compounds, all compounds are likely to be localized in the mitochondria except 3-tosylsedoheptulose, which localization is in the lysosome. A predicted plasma protein binding of 55.94%, 49.26%, 43.47% and 84.65% was recorded for 2-*O*-p-methylphenyl-1-thio-β-d-glucoside, 3-tosylsedoheptulose, 4-benzyloxy-6-hydroxymethyl-tetrahydropyran-2,3,5-triol and vitamin E respectively while −0.496, −1.017, −0.278 and 0.444 L/kg were their respective predicted volume distribution (Table 10). These four compounds were not predicted inhibitors and substrates of 1A2, 3A4, 2C9, 2C19 and 2D6 CYP_450_ isoforms. However, vitamin E was a predicted substrate for 2C9, 2C19 and 3A4 CYP_450_ isoforms while 3-tosylsedoheptulose was predicted to be a substrate for 2C9 CYP_450_ isoforms (Table 10). Clearance rate of 1.55, 0.724, 1.732 and 1.581 mL/min/kg was predicted for 2-*O*-p-methylphenyl-1-thio-β-d-glucoside, 3-tosylsedoheptulose, 4-benzyloxy-6-hydroxymethyl-tetrahydropyran-2,3,5-triol and vitamin E respectively while 0.98, 1.21, 0.93 and 1.95 h were their respective predicted half life (Table 10). The predicted toxicity profile of the compounds as tabulated in Table 10 revealed 2-*O*-p-methylphenyl-1-thio-β-d-glucoside, 3-tosylsedoheptulose, 4-benzyloxy-6-hydroxymethyl-tetrahydropyran-2,3,5-triol and vitamin E are non-mutagenic, non-carcinogen, non-skin sensitizers, non-inhibitors of human ether-a-go-go-related gene and classified as class III acute oral toxicity compounds. They have an LD_50_ of 736.03, 1146.95, 1967.05 and 1161.96 mg/kg respectively and meet food and drug administration (FDA) maximum recommended daily dose requirements.

## 3. Discussion

This study was carried out to identify potent DPP-IV antagonists from GC-MS identified phytocompounds in *N. latifolia* aqueous and ethanol leaf extracts. Development of DPP-IVi from plant-based sources is on the increase due to the observed side effects such as pancreatitis, cardiovascular challenges, renal- and hepatotoxicity of clinically approved DPP-4 inhibitors [20]. In silico approaches are cheap, proven and energy-saving techniques successfully applied in discovering leads from phytocompounds identified in different plant parts or large databases ensuring they succeed in the drug development process [32]. The phytoconstituents found in the ethanol and aqueous extract of *N. latifolia* belonged to various phytochemical classes such as alkaloids, phenolics, terpenes/terpenoids, fatty acid and others. Majority of these compounds were either phenolics or terpenes/terpenoids, which may be a major reason *N. latifolia* leaves have been verified to possess a wide range of pharmacological activities such as hepatoprotective, antimalarial and antifungal amongst others due to the synergistic action of these phytocompounds [31]. Notably, α-terpineol, γ-sitosterol, catechol, vitamin E, phytol, 9,12-octadecadienoic acid (Z,Z)- and dodecanoic acid are some bioactives that have been isolated and identified in various plants and fungi to illicit antioxidant, anti-inflammatory and antidiabetic activity by scavenging free radicals, preserving beta-cell function, ameliorating glucose-induced toxicity, attenuating oxidative and inflammatory stress [33,34,35,36,37,38]. SWISS-MODEL uses a modeling pipeline, which relies on OpenStructure comparative modeling engine to extract structural information of various templates to provide complete stoichiometry and the overall structure of the complex as inferred by homology modeling [39,40]. It generates a 3D protein model of the target sequence by extrapolating experimental information from an evolutionarily related protein structure that serves as a template [39]. The modeled DPP-IV protein was made up of two identical polypeptide chains, which had the same number, order and amino acid residues signifying a possible homo-dimer protein. The GMQE and QSQE score were also very high indicating structural similarity to the template as well as proper quaternary structure inter chain interaction. The QMEAN and Z-score also show the modeled protein behaves similar to experimentally determined DPP-IV pdb structures. Over 99.6% of the modeled protein was deemed to have local similarity to template indicating an overall quality structure. Nonetheless, the remaining 0.4% of the protein region (Ser288, Leu765 and Pro766) with low quality did not have any part to play in the binding site. The latter amino acid residue were located at the tail end of the protein. The modeled protein was adduced to be acidic because of a higher number D and E residues compared to N and Q composition with majorly alpha-helical and beta-sheet secondary conformations indicating structural orderliness of the protein as well as a low RMSD signifying a good model representation [14]. The residue distribution as visualized by the Ramachandran plot shows over 99% of the modeled 3D structure had high structural integrity with a usual feature due to the high overall G-factor not reaching the threshold of −0.5. The compatibility of the modeled protein with its amino acid sequence and the observed non-bond interactions among various atoms of the modeled protein signified the model had good stereochemical quality [41]. This further corroborates the highly reliable quality of the modeled structure. The prediction of a drug targets active site with high substrate affinity is an important phase in the discovery of therapeutic compounds as they improve the clinical progression of compounds amidst the various uncertainties surrounding pocket estimation [42]. Most servers and software are predicting drug target active sites use one unique pocket estimation method. It is always imperative to use two or more to validate the prediction. In DoGSiteScorer, a grid-based method incorporating Gaussian filter difference is used to detect potential binding pockets thereafter splitting them into subpockets based the protein 3D structure. A support vector machine, which has a subset of meaningful descriptors integrated is used to predict druggability scores of the pockets and subpockets [43]. PockDrug detects druggable binding pocket through ligand proximity via several thresholds and amino atoms present at the binding site surface [44]. Despite the differences in the mode of cavity prediction by these two servers, the binding site as detected by DoGSiteScorer was validated by PockDrug with an identical cavity location. This step always precedes docking simulations as docking is simulated to predict the interaction between therapeutic ligands or substrates in the binding pocket of their target macromolecule [45]. Like drug target active sites prediction, two or more docking servers or software’s will ensure there is an elimination of false negatives and positives. iGEMDOCK and AutoDock Vina are software that predicts binding modes between ligands and targets through an evolutionary approach in empirical scoring function [46] and sophisticated gradient optimization method in local optimization procedure [47] respectively. Comparing respective iGEMDOCK and AutoDock Vina binding values, 2-*O*-p-methylphenyl-1-thio-β-d-glucoside, 3-tosylsedoheptulose, 4-benzyloxy-6-hydroxymethyl-tetrahydropyran-2,3,5-triol and vitamin E exhibited considerable good binding mode with DPP-IV compared with saxagliptin and alogliptin, which are some United States food and drug administration (USFDA) approved DPP-IVi. Saxagliptin and alogliptin are two different DPP-IVi that belong to peptide and non-peptide mimetic class respectively based on their structure while they also belong to class 1 and 2 inhibitors respectively based on inhibitory action [19]. Kalhotra et al. [32] reported the hydrophobic S1, S2, N-terminal recognition and catalytic triad clefts as the four important domains responsible for DPP-IV inhibitory activity. From this study, 2-*O*-p-methylphenyl-1-thio-β-d-glucoside could be inferred to have a class 1 binding mode due to its simulated interaction with hydrophobic S1, S2 amino acid residues and hydrogen bond formation with Ser630 [48]. The modeled binding mode was similar to saxagliptin as Glu205 and Glu206 interaction ensures both 2-*O*-p-methylphenyl-1-thio-β-d-glucoside and saxagliptin are aligned in the β-propeller region such that there is a covalent interaction with one or more amino acid residues (Ser630, His740 and Asp708) in the catalytic triad domain [49]. 3-tosylsedoheptulose had some interactions similar to that of 2-*O*-p-methylphenyl-1-thio-β-d-glucoside, but the π-π interaction with Tyr547 may suggest a conformational change in the S1ʹ domain mimicking a class 2 binding model [50,51]. 4-Benzyloxy-6-hydroxymethyl-tetrahydropyran-2,3,5-triol formed various interactions with the β-propeller, S1 and S2 domain with a π–π interaction with Tyr547 amino residue that would propose a class 2 binding model as a result of a possible S1ʹ domain conformational change [50,51]. For vitamin E simulated binding, common β-propeller, S1 and S2 domain interactions were present, but interestingly with π-π and Van der Waals interaction with Phe357 and Ser209 amino residues is suggestive of binding with the S2 extensive or S3 domain. This domain has been reported to be larger in DPP-IV compared to other DPP isoforms, thereby having the ability to accommodate the large hydrophobic phytyl side chain of vitamin E suggesting class 3 binding [50]. Since these hit compounds exhibited lower binding energies and bound tightly to DPP-IV, they may play a role in preventing the rapid degradation of GLP1 and GIP, concomitantly having a remarkable impact on glycemic homeostasis and ultimately diabetes [8]. The binding classification of these lead compounds has not been previously reported to the best of our knowledge. It is worthy of note that a similar report was generated for oxidovanadium complexes in the binding pocket of DPP-IV [51]. Meduru et al. [52], have previously reported a positive correlation between predicted binding score and experimental activity values suggesting the predicted low binding affinity of 2-*O*-p-methylphenyl-1-thio-β-d-glucoside, 3-tosyl sedoheptulose, 4-benzyloxy-6-hydroxymethyl-tetrahydropyran-2,3,5-triol and vitamin E on DPP-IV may be translated to low inhibitory concentrations (IC_50_) when validated experimentally. Prediction of physicochemical and ADMET properties are a cheap and time-saving alternative in developing therapeutic leads as compared to in vivo testing. This process helps eliminate compounds with both poor pharmacokinetic characteristics and high toxicity in biological systems leading to failed drug development [53]. Lipinski RO5 states the molecular weight, octanol-water coefficient, hydrogen bond acceptors and donors of the compounds should be no more than 500, 5, 10 and 5 g/mol respectively. However, a consensus is allowed if only one parameter is violated by the compound [27,54]. All the compounds passed the RO5 suggesting they will likely be orally active drugs [54,55]. TPSA and LogP are important parameters that determine how compounds are absorbed [56]. Vitamin E had the highest LogP and lowest TPSA while 3-tosylsedoheptulose had the lowest LogP and highest TPSA. Due to these physicochemical properties, all compounds bar 3-tosylsedoheptulose were able to permeate the blood brain barrier (BBB) with vitamin E being the most permeable. Thus, these compounds could also have neurodegenerative therapeutic applications. As regards Caco-2 permeability and human intestinal absorption, only vitamin E was positive suggesting easy absorption of this compound in the brush of the intestinal wall [57]. The other leads could, however, make use of carrier-mediated transport due to their lower lipophilic physicochemical properties compared to vitamin E [58]. Since the identified leads were neither substrates nor inhibitors of P-glycoprotein, renal organic cation transporter, human ether-a-go-go-related gene blockers, CYP1A2 and CYP2D6 as well as low CYP inhibitory promiscuity, there is a promising probability not to cause a multidrug resistance phenomenon, drug metabolism malfunctioning and toxicity elevation [45,59]. In addition, the risk of CYP drug interactions via the various isoforms found in different body tissues is greatly reduced as non-inhibitors of various CYP_450_ isoforms do not impede the biotransformation of CYP_450_ metabolized drugs [56,60]. Compounds that inhibit CYP_450_ not only decrease the enzymatic activity but lead to the accumulation of drugs to toxic levels [61]. These compounds do not show carcinogenicity, mutagenicity, skin sensitivity and oral toxicity. They meet the maximum recommended daily dose of USFDA with a high LD_50_ suggesting the compounds are non-toxic and have a low chance of causing toxicity.

## 4. Materials and Methods

### 4.1. Chemicals and Reagents 

Ethanol used was of analytical grade and obtained from BDH chemicals, Poole, England.

### 4.2. Plant Collection, Identification and Extract Preparation

*Nauclea latifolia* leaves were sourced from farms in Oyo State, Nigeria, in November 2016, identified by Dr. J. O. Popoola and deposited in the herbarium of Biological Sciences Department of Covenant University, Ota, Ogun State with herbarium number NL/CUBio/H810. The aqueous and ethanol extract of *N. latifolia* (NLA and NLE) leaves were prepared as reported by Iheagwam et al. [62].

### 4.3. GC-MS Analysis

NLA and NLE were subjected to GC-MS analysis using GCMS-QP2010SE SHIMADZU JAPAN with a fused Optima-5MS capillary column of 30 m length, 0.25 mm diameter and 0.25 µm film thickness. GC conditions: Pure helium (1.56 mL/min flow rate and 37 cm/s linear velocity); Injector temperature (200 °C); column oven temperature (initially 60 °C increased to 160 °C then 250 °C at 10 °C/min with 2 min/increment hold time); injection volume and split ratio (0.5 μL and 1:1 respectively). MS conditions: Ion source and interface temperature (230 °C and 250 °C respectively), solvent delay (4.5 min) recorded in a scan range of 50–700 amu. Unknown constituents were identified by comparing the retention time, mass spectral data and fragmentation pattern of the extracts with established libraries (National Institute of Standards and Technology (NIST) and Wiley libraries) [63].

### 4.4. Ligand Modeling 

The identified structures were downloaded from PubChem (Available online: https://pubchem.ncbi.nlm.nih.gov/) as .sdf files. The files were converted to their corresponding three-dimensional (3D) structures in protein data bank (PDB) format using Open Babel v2.3.2 [64]. LigParGen [65] was used to generate optimized potentials for liquid simulations (OPLS) force fields for the ligands. Non-polar hydrogens and Gasteiger charges were merged and added respectively using AutoDock4.2 [66,67].

### 4.5. Protein Sequence Retrieval, Model Optimization and Energy Minimization

DPP-IV sequence was obtained from NCBI (Available online: https://www.ncbi.nlm.nih.gov/protein) through NCBI reference sequence identification number (NP_001926.2). The obtained sequence was queried using the basic local alignment search tool (BLASTp) against the PDB (Available online: http://www.pdb.org) program to identify related protein structural templates [68]. The homology modeling approach was performed via the SWISS-MODEL to generate an optimized model [39]. The 3D structure of DPP-IV was modeled based on a deposited crystal structure of DPP-IV from *Homo sapiens* with high resolution, sequence identity, domain coverage and E-value after blasting. The modeled DPP-IV structure energy level was minimized using 3Drefine [69].

### 4.6. Model Evaluation, Physicochemical Analysis and Pocket Identification

The structural chemistry stability of the minimized DPP-IV protein was evaluated using PROCHECK [70], ERRAT [71] and Verify_3D [72] while the physicochemical properties were analyzed using ProtoParam [73]. DoGSiteScorer [43] was used to identify possible active druggable binding sites of the protein with further verification using PockDrug [44].

### 4.7. Molecular Docking Simulation

Molecular docking (MD) of the modeled ligands in the binding pocket of DPP-IV was carried out using iGEMDOCK v2.1 [46]. A population size of 300, 80 generations and 10 solutions were the parameters assigned to screen the ligands in comparison with clinically approved DPP-IVi (Saxagliptin and Vildagliptin). To further validate results and avoid false positives, ligands were also docked using AutoDock Vina [47]. Prior to AutoDock Vina docking simulations, Autodock 4.2 [66,67] was used to compute Gasteiger charges, assign nonpolar hydrogen and set the grid map at 18 × 24 × 26 spaced at 1 Å. 

### 4.8. Druglikeness, Pharmacokinetic and Toxicity Prediction

Lipinski rule of five (RO5) [27], ADMETlab [74] and admetSAR [75] were used to predict various drug-likeness, pharmacokinetic and toxicity parameters respectively of the docked ligands that had a better binding fit than the clinically approved DPP-IVi.

## 5. Conclusions

This study has identified 50 phytoconstituents present in *N. latifolia* leaf extracts using GC-MS proving the therapeutic potential of this plant against diabetes and other ailments. However, only four phytocompounds have been identified to possess comparable binding score with two clinically prescribed DPP-IVi as well as exhibit promising ADMET properties. This to the best of our knowledge is the first time, the binding classification of these four phytocompounds have been reported. Nonetheless, vitamin E concentration should be given close attention to during the lead optimization process due to its high permeability of the BBB to avoid eliciting adverse effect and CYP substrate ability. Molecular mechanics energies combined with the Poisson-Boltzmann or generalized Born and surface area continuum solvation (MM-PBSA/GBSA) and molecular dynamics analyses are required to corroborate these results. Further in vitro and in vivo research can be done to confirm and validate the pharmacological significance of these lead compounds for further development as potent DPP-IV antagonistic drugs.

## Figures and Tables

**Figure 1 ijms-20-05913-f001:**
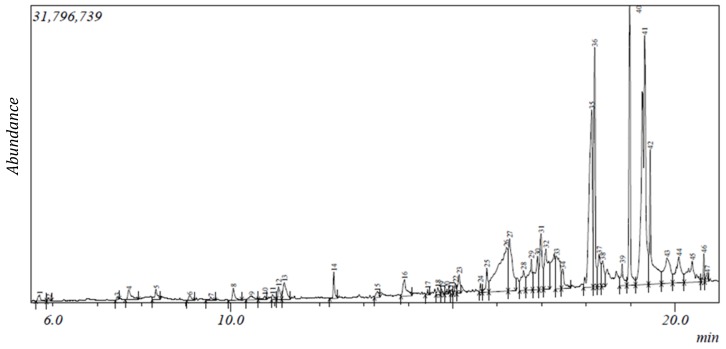
Gas chromatogram of NLE.

**Figure 2 ijms-20-05913-f002:**
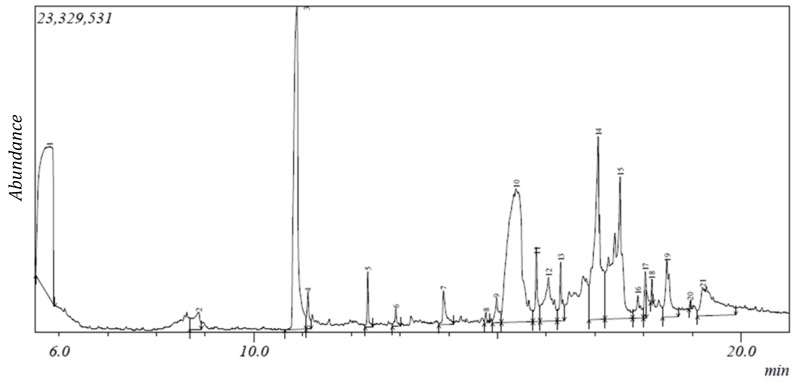
Gas chromatogram of NLA.

**Figure 3 ijms-20-05913-f003:**
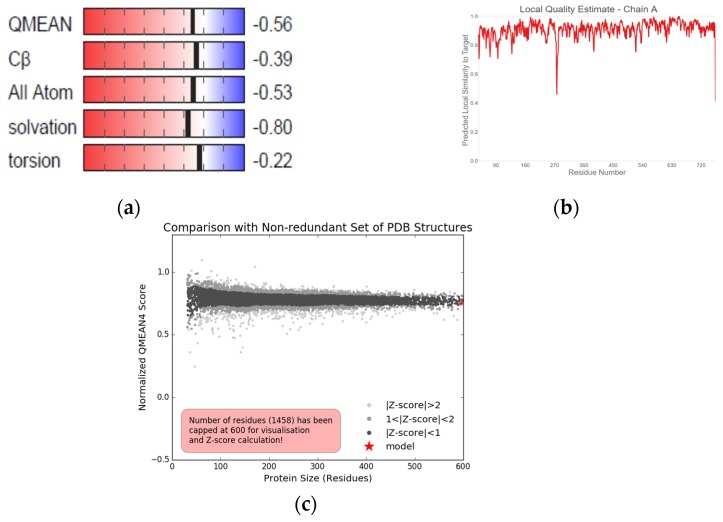
The (**a**) global quality estimate makeup, (**b**) local quality estimate and (**c**) comparison plots of modeled DPP-IV.

**Figure 4 ijms-20-05913-f004:**
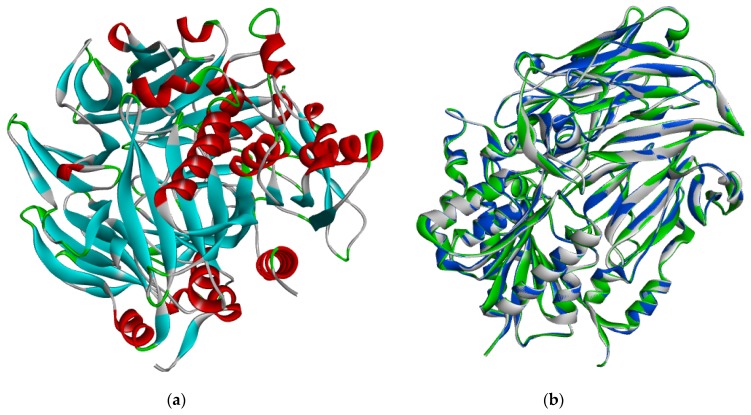
Cartoon representations of (**a**) 3D modeled *Homo sapiens* dipeptidyl peptidase IV structure showing α helices (red), β sheet (blue) and loops (green). (**b**) 3D structural superimposition of 1wcy (blue), modeled DPP-IV (grey) and energy minimized DPP-IV (green).

**Figure 5 ijms-20-05913-f005:**
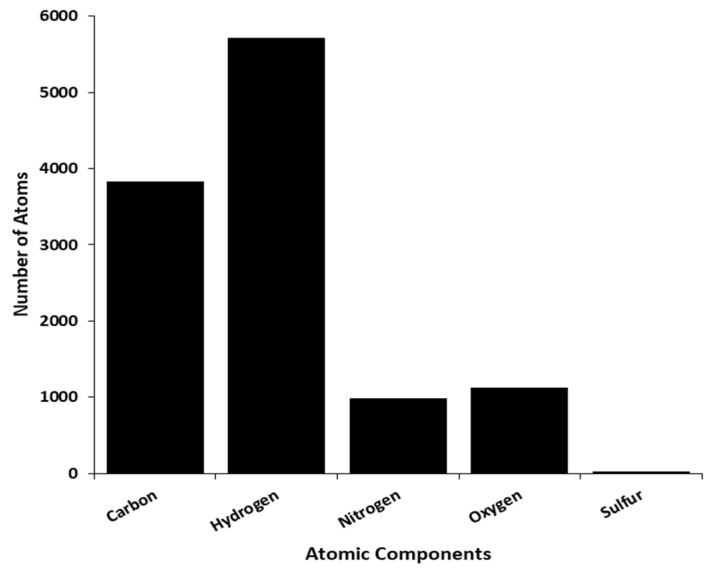
Representation of the DPP-IV atomic composition.

**Figure 6 ijms-20-05913-f006:**
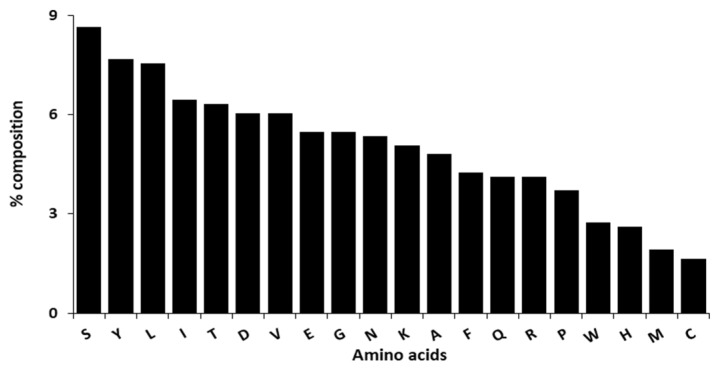
Representation of the DPP-IV amino acid residues composition.

**Figure 7 ijms-20-05913-f007:**
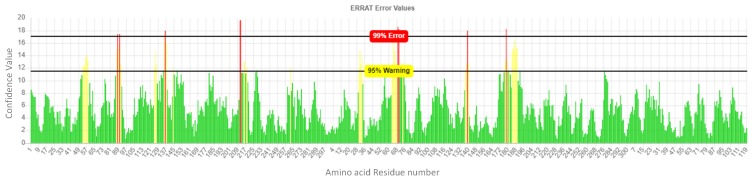
Quality factor plot of the minimized modeled DPP-IV.

**Figure 8 ijms-20-05913-f008:**
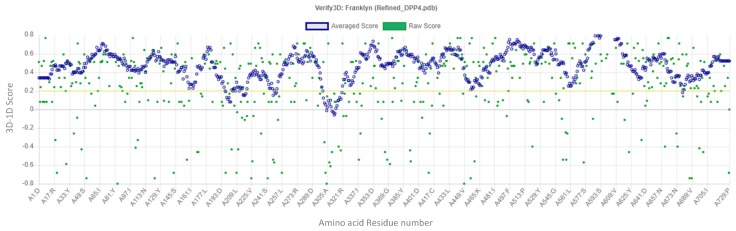
3D verification plot of the minimized modeled DPP-IV structure.

**Figure 9 ijms-20-05913-f009:**
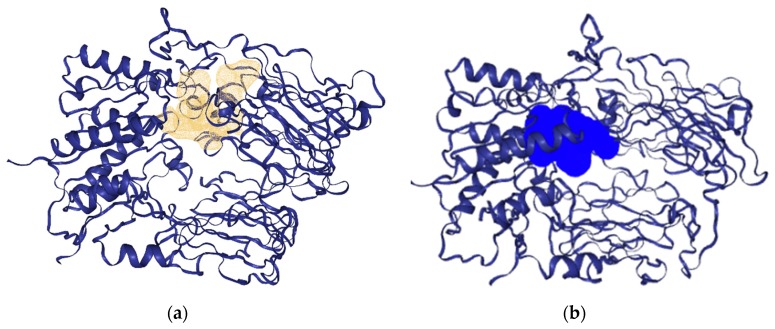
Identified DPP-IV binding pocket as simulated by (**a**) DoGSiteScorer in gold and (**b**) PockDrug in blue.

**Figure 10 ijms-20-05913-f010:**
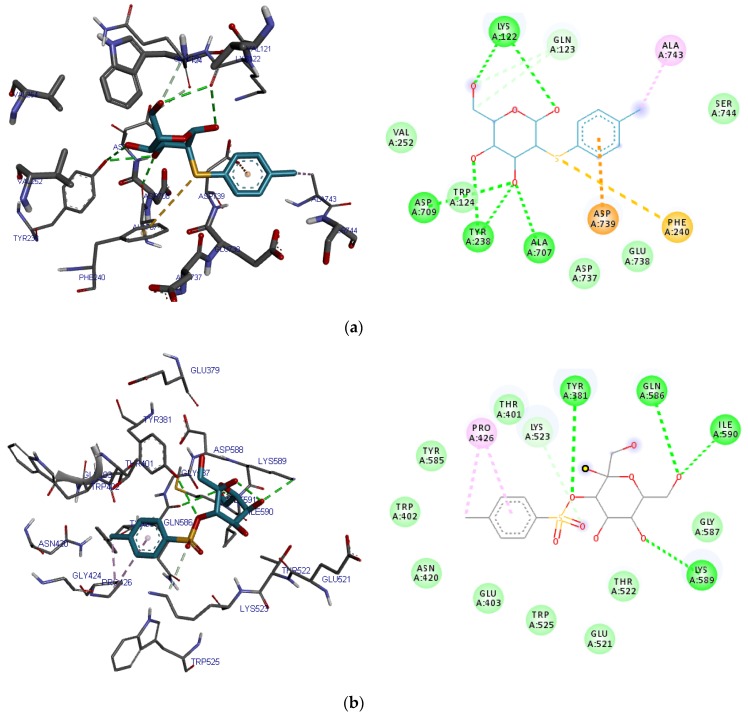
3D and 2D binding poses of (**a**) 2-*O*-p-methylphenyl-1-thio-β-d-glucoside, (**b**) 3-tosylsedoheptulose, (**c**) 4-benzyloxy-6-hydroxymethyl-tetrahydropyran-2,3,5-triol, (**d**) vitamin E, (**e**) alogliptin and (**f**) saxagliptin simulated by iGEMDOCK. For each ligand, hydrogen, carbon-hydrogen, unfavorable and π–π bonds are depicted as green, light blue, red and any other colored (purple, magenta, orange and yellow) broken lines respectively.

**Figure 11 ijms-20-05913-f011:**
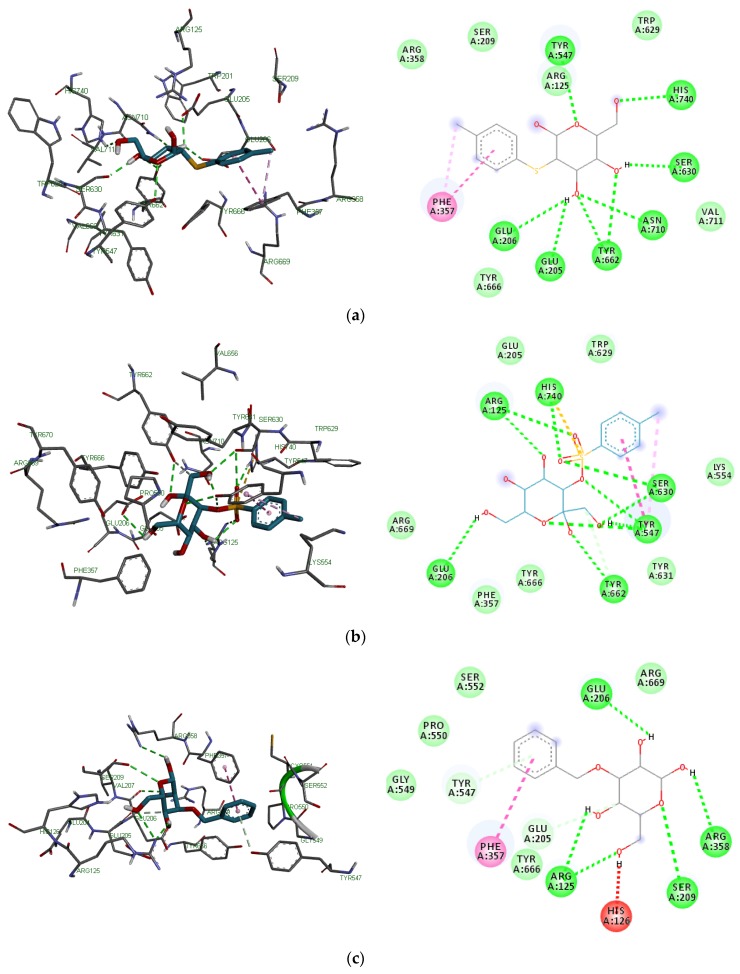
3D and 2D binding poses of (**a**) 2-*O*-p-methylphenyl-1-thio-β-d-glucoside, (**b**) 3-tosylsedoheptulose, (**c**) 4-benzyloxy-6-hydroxymethyl-tetrahydropyran-2,3,5-triol, (**d**) vitamin E, (**e**) alogliptin and (**f**) saxagliptin simulated by AutoDock Vina. For each ligand, hydrogen, carbon-hydrogen, unfavorable and π–π bonds are depicted as green, light blue, red and other colored (purple and magenta) broken lines respectively.

**Table 1 ijms-20-05913-t001:** Gas chromatography-mass spectroscopy (GC-MS) identified phytocompounds in NLE.

S/N	Compound	Retention Time (min)	Area (%)	Formula	Molecular Weight	Compound Classification
1	2-Furanmethanediol, dipropionate	5.702	0.31	C_11_H_14_O_5_	226	fatty acid
2	2-Furanmethanol	5.902	0.09	C_5_H_6_O_2_	98	alcohol
3	2-Oxopentanedioic acid	7.45	0.09	C_5_H_6_O_5_	146	carboxylic acid
4	1,3-Cyclohexanedione	7.708	0.71	C_6_H_8_O_2_	112	phenolic
5	4-Benzyloxy-6-hydroxymethyl-tetrahydropyran-2,3,5-triol	8.322	0.61	C_13_H_18_O_6_	270	phenolic
6	1H-Azonine, octahydro-1-nitroso-	9.081	0.27	C_8_H_16_N_2_O	156	alkaloid
7	Phenylethyl alcohol	9.556	0.19			phenolic
8	4H-Pyran-4-one,2,3-dihydro-3,5-dihydroxy-6-methyl- ^#^	10.065	0.54	C_6_H_8_O_4_	144	phenolic
9	Benzoic acid	10.476	0.41	C_7_H_6_O_2_	122	phenolic
10	α-Terpineol *^@^	10.775	0.2	C_10_H_18_O	154	terpene
11	1-(Methoxymethoxy)-3-methyl-3-hydroxybutane	10.947	0.18	C_7_H_16_O_3_	148	alcohol
12	Benzofuran, 2,3-dihydro-	11.083	0.53	C_8_H_8_O	120	phenolic
13	5-Hydroxymethylfurfural	11.213	1.13	C_6_H_6_O_3_	126	carbohydrate
14	2-Methoxy-4-vinylphenol	12.324	0.73	C_9_H_10_O_2_	150	phenolic
15	1,2,4-Benzenetriol *	13.303	0.4	C_6_H_6_O_3_	126	phenolic
16	2-Hydroxy-5-methylisophthalaldehyde	13.916	1.25	C_9_H_8_O_3_	164	phenolic
17	3-Tosylsedoheptulose	14.434	0.12	C_14_H_20_O_9_S	364	carbohydrate
18	Caprylic anhydride	14.667	0.25	C_16_H_30_O_3_	270	fatty acid
19	4,4-Dimethyl-cyclohex-2-en-1-ol	14.748	0.14	C_8_H_14_O	126	phenolic
20	5-Caranol, trans, trans-(+)-	14.868	0.14	C_10_H_18_O	154	terpenoid
21	11-(2-Cyclopenten-1-yl) undecanoicacid, (+)-	14.981	0.11	C_16_H_28_O_2_	252	fatty acid
22	4-Hydroxy-2-hydroxyaminopyrimidine	15.062	0.35	C_4_H_5_N_3_O_2_	127	alkaloid
23	9-Oxabicyclo[3.3.1]nonane-2,6-diol	15.143	0.77	C_8_H_14_O_3_	158	phenolic
24	Megastigmatrienone	15.627	0.23	C_13_H_18_O	190	terpene
25	2-Cyclohexen-1-one, 4-(3-hydroxy-1-butenyl)-3,5,5-trimethyl-, [R-[R*,R*-(*E*)]]-	15.768	0.97	C_13_H_20_O_2_	208	terpene
26	1,2,3,5-Cyclohexanetetrol, (1-alpha,2-beta,3-alpha,5-beta)-	16.278	3.64	C_6_H_12_O_4_	148	phenolic
27	Tridecanoic acid	16.592	1.38	C_13_H_26_O_2_	214	fatty acid
28	Dodecanoic acid *	16.77	2.09	C_12_H_24_O_2_	200	fatty acid
29	[1,1’-Bicyclopropyl]-2-octanoic acid, 2’-hexyl-, methyl ester	16.908	2.22	C_21_H_38_O_2_	322	fatty acid ester
30	2-*O*-p-Methylphenyl-1-thio-β-d-glucoside	16.984	2.47	C_13_H_18_O_5_S	286	carbohydrate
31	3-*O*-Methyl-d-glucose	17.333	2.68	C_7_H_14_O_6_	194	carbohydrate
32	*n*-Hexadecanoic acid ^#^*	18.126	10.86	C_16_H_32_O_2_	256	fatty acid
33	Hexadecanoic acid, ethyl ester *	18.195	7.32	C_18_H_36_O_2_	284	fatty acid ester
34	γ-Sitosterol *^@^	18.375	1.18	C_29_H_50_O	414	terpenoid
35	9,9-Dimethoxybicyclo[3.3.1]nona-2,4-dione	18.814	1.04	C_11_H_16_O_4_	212	phenolic
36	Phytol ^#@^	18.983	7.24	C_20_H_40_O	296	terpenoid
37	Ethyl Oleate	19.319	18	C_20_H_38_O_2_	310	fatty acid ester
38	Octadecanoic acid, ethyl ester ^#^	19.441	5.51	C_20_H_40_O_2_	312	fatty acid ester
39	Androstan-17-one, 16,16-dimethyl-(5-alpha)-	19.824	3.05	C_21_H_34_O	302	terpenoid
40	1-Naphthalenol,decahydro-1,4a-dimethyl-7-(1-methylethylidene)-	20.088	2.69	C_15_H_26_O	222	phenolic
41	17-Octadecynoic acid	20.721	0.29	C_11_H_21_N	280	fatty acid

* Compounds with antioxidant activity; ^#^ compounds with anti-inflammatory activity; ^@^ compounds with antidiabetic activity; *^#@^ Source: Dr Duke’s: Phytochemical and ethnobotanical databases.

**Table 2 ijms-20-05913-t002:** GC-MS identified phytocompounds in NLA.

S/N	Compound	Retention Time (min)	Area (%)	Formula	Molecular Weight	Compound Classification
1	2,3-Butanediol	5.805	20.04	C_4_H_10_O_2_	90	alcohol
2	2,5-Dimethyl-4-hydroxy-3(2H)-furanone	8.869	1.32	C_6_H_8_O_3_	128	phenolic
3	Catechol *	10.886	14.84	C_6_H_6_O_2_	110	phenolic
4	Benzofuran, 2,3-dihydro-	11.107	0.74	C_8_H_8_O	120	phenolic
5	2-Methoxy-4-vinylphenol	12.339	0.81	C_9_H_10_O_2_	150	phenolic
6	2,7-Octadiene-1,6-diol, 2,6-dimethyl-	12.914	0.38	C_10_H_18_O_2_	170	terpene
7	2-Hydroxy-5-methylisophthalaldehyde	13.893	1.16	C_9_H_8_O_3_	164	phenolic
8	Bicyclo[2.2.1]heptan-2-one, 1-(bromomethyl)-7,7-dimethyl-,(1S)-	14.764	0.21	C_10_H_15_BrO	230	phenolic
9	11-(2-Cyclopenten-1-yl)undecanoic acid, (+)-	14.982	0.9	C_16_H_28_O_2_	252	fatty acid
10	9-Oxabicyclo[3.3.1]nonane-2,6-diol	15.384	19.75	C_8_H_14_O_3_	158	phenolic
11	2-Cyclohexen-1-one, 4-(3-hydroxy-1-butenyl)-3,5,5-trimethyl-, [R-[R*,R*-(*E*)]]-	15.807	1.67	C_13_H_20_O_2_	208	terpene
12	9,9-Dimethoxybicyclo[3.3.1]nona-2,4-dione	16.052	3.44	C_11_H_16_O_4_	212	phenolic
13	5,5,8a-Trimethyl-3,5,6,7,8,8a-hexahydro-2H-chromene	17.07	11.13	C_12_H_20_O	180	phenolic
14	[1,1’-Bicyclopropyl]-2-octanoic acid, 2’-hexyl-, methyl ester	17.522	12.01	C_21_H_38_O_2_	322	fatty acid ester
15	*n*-Hexadecanoic acid ^#^*	18.035	0.89	C_16_H_32_O_2_	256	fatty acid
16	Ethyl 14-methyl-hexadecanoate	18.171	0.23	C_19_H_38_O_2_	298	fatty acid
17	Vitamin E ^#^*	18.481	2.89	C_22_H_30_O_5_	430	terpenoid
18	Phytol ^#@^	18.96	0.07	C_20_H_40_O	296	terpenoid
19	9,12-Octadecadienoic acid (Z,Z)- ^#@^	19.217	5.03	C_18_H_32_O_2_	280	fatty acid

* Compounds with antioxidant activity; ^#^ compounds with anti-inflammatory activity; ^@^ compounds with antidiabetic activity; *^#@^ Source: Dr Duke’s: Phytochemical and ethnobotanical databases.

**Table 3 ijms-20-05913-t003:** Homology modeling template results.

S/N	Template	GMQE	QSQE	Sequence Identity	Sequence Similarity	Resolution	Oligomeric State
1	1wcy	0.99	1	100	0.62	2.2 Å	homo-dimer
2	3qbj	0.99	1	99.73	0.62	2.2 Å	homo-dimer
3	2qt9	0.99	1	99.87	0.62	2.1 Å	homo-dimer
4	2bgr	0.99	0.96	100	0.62	2.0 Å	homo-dimer
5	5lls	0.94	1	88.38	0.59	2.4 Å	homo-dimer
6	2gbg	0.94	1	84.99	0.58	3.0 Å	homo-dimer
7	2jid	0.99	0.93	100	0.62	2.8 Å	homo-dimer
8	1orv	0.97	0.68	88.19	0.59	1.8 Å	homo-tetramer
9	3f8s	0.99	-	100	0.62	2.4 Å	monomer
10	5vta	0.94	-	85.01	0.58	2.8 Å	hetero-trimer
11	4ffv	0.93	1	84.99	0.58	2.4 Å	hetero-hexamer

GMQE: Global model quality estimation; QSQE: Quaternary structure quality estimate.

**Table 4 ijms-20-05913-t004:** Generated energy-minimized models using 3D refine.

Model No	3D Refine Score	RWplus	MolProbity
**5**	31,326.3	−174,719.07	1.412
**4**	31,591.2	−174,480.02	1.302
**3**	31,993.4	−174,208.07	1.344
**2**	32,698.8	−174,067.52	1.258
**1**	35,422.7	−174,035.11	1.190

**Table 5 ijms-20-05913-t005:** Structural evaluation of energy minimized modeled DPP-IV, modeled DPP-IV and template.

	PROCHECK				G-Factor ^2^		
	Most Favored (%)	Additional Allowed (%)	Generously Allowed (%)	Disallowed (%)	Torsion Angles	Covalent Geometry	Overall Average
DPP4.A	89.6	10.1	0.2	0.2	−0.20	0.11	−0.07
Min DPP4	88.8	10.6	0.3	0.3	−0.02	−1.07	−0.52
1wcy.A	86.7	12.9	0.3	0.2	0.13	0.54	0.30

2: Degree (°).

**Table 6 ijms-20-05913-t006:** Docking results of ligands and standard drugs on dipeptidyl peptidase IV.

		Docking Score (kcal/mol)				
S/N	Compound	IGEMDOCK				AutoDock Vina
		TE	VdW	HB	Elec	BE
1	1,2,3,5-Cyclohexanetetrol, 1-alpha,2-beta,3-alpha,5-beta-	−64.14	−32.47	−31.67	0.00	−5.2
2	1,2,4-Benzenetriol	−65.06	−43.51	−21.54	0.00	−4.9
3	1,3-Cyclohexanedione	−58.68	−43.60	−15.08	0.00	−4.3
4	1-(Methoxymethoxy)-3-methyl-3-hydroxybutane	−67.65	−51.15	−16.50	0.00	−4.3
5	1-Naphthalenol,decahydro-1,4a-dimethyl-7-(1-methylethylidene)	−64.18	−58.18	−6.00	0.00	−6.5
6	11-(2-Cyclopenten-1-yl)undecanoicacid, (+)-	−70.60	−63.76	−8.77	1.93	−5.1
7	17-Octadecynoic acid	−73.26	−63.31	−9.89	−0.06	−4.9
8	1H-Azonine, octahydro-1-nitroso-	−54.18	−41.95	−12.23	0.00	−4.7
9	2,3-Butanediol	−51.07	−36.18	−14.89	0.00	−4.3
10	2,5-Dimethyl-4-hydroxy-3(2H)-furanone	−56.19	−44.76	−11.43	0.00	−4.8
11	2,7-Octadiene-1,6-diol, 2,6-dimethyl-	−62.21	−52.88	−9.33	0.00	−4.8
12	2-Cyclohexen-1-one, 4-(3-hydroxy-1-butenyl)-3,5,5-trimethyl-, [R-[R*,R*-(*E*)]]-	−61.02	−55.77	−5.25	0.00	−6.1
13	2-Furanmethanediol, dipropionate	−68.62	−59.32	−9.30	0.00	−5.7
14	2-Furanmethanol	−56.91	−37.00	−19.91	0.00	−4.4
15	2-Hydroxy-5-methylisophthalaldehyde	−71.18	−55.30	−15.88	0.00	−5.1
16	2-Methoxy-4-vinylphenol	−70.07	−54.15	−15.92	0.00	−5.1
17	2-*O*-p-Methylphenyl-1-thio-β-d-glucoside	−76.67	−60.32	−16.35	0.00	−6.9
18	2-Oxopentanedioic acid	−64.36	−39.96	−20.20	−4.20	−5.2
19	3-*O*-Methyl-d-Glucose	−79.56	−48.45	−31.12	0.00	−5
20	3-Tosylsedoheptulose	−79.45	−64.19	−15.26	0.00	−7
21	4,4-Dimethyl-cyclohex-2-en-1-ol	−51.95	−39.09	−12.87	0.00	−4.6
22	4-Benzyloxy-6-hydroxymethyl-tetrahydropyran-2,3,5-triol	−90.22	−59.82	−30.40	0.00	−6.8
23	4-Hydroxy-2-hydroxyaminopyrimidine	−72.29	−47.82	−24.47	0.00	−5.5
24	4H-Pyran-4-one,2,3-dihydro-3,5-dihydroxy-6-methyl-	−73.41	−44.81	−28.61	0.00	−5.3
25	5,5,8a-Trimethyl-3,5,6,7,8,8a-hexahydro-2H-chromene	−59.09	−55.59	−3.50	0.00	−5.6
26	5-Caranol, trans, trans-(+)-	−57.52	−44.52	−13.00	0.00	−5.1
27	5-Hydroxymethylfurfural	−72.97	−57.68	−15.29	0.00	−5.1
28	9,12-Octadecadienoic acid (Z,Z)-	−72.42	−68.72	−2.34	−1.37	−5.2
29	9,9-Dimethoxybicyclo[3.3.1]nona-2,4-dione	−66.88	−52.88	−14.00	0.00	−4.8
30	9-Oxabicyclo[3.3.1]nonane-2,6-diol	−69.24	−49.12	−20.12	0.00	−4.8
31	[1,1’-Bicyclopropyl]-2-octanoic acid, 2’-hexyl-, methyl ester	−70.68	−67.18	−3.50	0.00	−6
32	α-Terpineol	−63.97	−58.97	−5.00	0.00	−5.2
33	Androstan-17-one, 16, 16-dimethyl-(5.alpha.)-	−63.14	−60.74	−2.40	0.00	−7.3
34	Benzofuran, 2,3-dihydro-	−54.13	−44.07	−10.06	0.00	−4.7
35	Benzoic acid	−56.65	−47.63	−9.23	0.21	−5
36	Bicyclo[2.2.1]heptan-2-one, 1-(bromomethyl)-7,7-dimethyl-,(1S)-	−56.79	−46.29	−10.50	0.00	
37	Caprylic anhydride	−65.63	−56.09	−9.54	0.00	−4.6
38	Cathecol	−65.43	−43.47	−21.95	0.00	−4.4
39	Dodecanoicacid	−50.00	−45.96	−4.04	0.00	−4.6
40	Ethyl 14-methyl-hexadecanoate	−66.40	−62.90	−3.50	0.00	−5.1
41	Ethyl Oleate	−74.10	−69.11	−4.99	0.00	−5.4
42	Hexadecanoic acid, ethyl ester	−63.14	−59.64	−3.50	0.00	−4.5
43	*n*-Hexadecanoic acid	−71.33	−59.08	−10.50	−1.75	−4.4
44	Megastigmatrienone	−62.48	−58.82	−3.66	0.00	−6.2
45	Octadecanoic acid, ethyl ester	−64.67	−61.23	−3.44	0.00	−4.9
46	Phenylethyl Alcohol	−61.00	−47.87	−13.14	0.00	−4.7
47	Phytol	−70.24	−66.74	−3.50	0.00	−5.6
48	Tridecanoic Acid	−60.93	−58.06	−3.50	0.62	−4.5
49	Vitamin E	−77.10	−67.80	−9.30	0.00	−6.7
50	γ-Sitosterol	−69.45	−65.95	−3.50	0.00	−7.8
51	Alogliptin	−82.55	−63.39	−19.17	0.00	−6.7
52	Saxagliptin	−76.74	−51.85	−24.90	0.00	−6.7

TE = total energy, VDW = Van der Waals interaction, HB = hydrogen bond, Elec = electrostatic interaction, BE = binding Energy.

**Table 7 ijms-20-05913-t007:** Interacting amino acid residues stabilizing ligands in the DPP-IV catalytic site simulated by iGEMDOCK.

Compound	Hb	CHb	VdW	π-π
2-*O*-p-Methylphenyl-1-thio-β-d-glucoside	Lys122, Asp709, Ala707, Tyr238	Gln123	Val252, Ser744, Trp124, Asp737, Glu738	Asp739, Phe240, Ala743
3-Tosylsedoheptulose	Tyr381, Gln 586, Lys 598, Ile 590	Lys523	Thr401, Trp402, Glu403, Asn420, Glu521, Thr522, Trp525, Tyr585, Gly587	Pro426
4-Benzyloxy-6-hydroxymethyl-tetrahydropyran-2,3,5-triol	Lys122, Gly741, Asp739, His740	Arg125	Gln123, Trp124, Ser630, Asp709, Asn710, Ala743	Trp629, His740
Vitamin E	Lys392, Asp393, Cys394		Thr350, Thr351, Ser349, Ser376, Glu378, Glu347, His592, Asp588, Phe387, Asn377, Met348, Ile346	Ile375, Cys385, 394, Phe396, Lys392
Saxagliptin	Lys175, Ile148, Asn150, Ser182		Tyr166, Glu146, Pro149, 181,	Arg147
Alogliptin	His740, Trp629, Lys554	Ser630	Gly628, Gly632, Gly741, Val546, Trp627, Tyr752	Trp629, Tyr547

Hb: hydrogen bond; CHb: carbon hydrogen bond; VdW: Van der Waals interaction; π–π: pi–pi bond.

**Table 8 ijms-20-05913-t008:** Interacting amino acid residues stabilizing hit ligands in the DPP-IV catalytic site simulated by AutoDock Vina.

Compound	Hb	CHb	VdW	π-π
2-*O*-p-Methylphenyl-1-thio-β-d-glucoside	Tyr547, Tyr662, Glu205, Glu206, Asn710, Ser630, His740		Arg125, Arg358, Ser209, Trp629, Val711, Tyr666	Phe357
3-Tosylsedoheptulose	Glu206, Tyr662, Tyr547, Ser630, Arg125, His740		Arg669, Phe357, Tyr666, Tyr631, Lys554, Trp629, Glu205	His740, Tyr547
4-Benzyloxy-6-hydroxymethyl-tetrahydropyran-2,3,5-triol	Arg125, Arg358, Ser209, Glu206	Glu205	Pro550, Ser552, Gly549, Arg669, Tyr666,	Tyr547, Phe357
Vitamin E	Glu206		His740, Trp629, Ser209, Ser552, Ser630, Gln553, Arg358, Arg669, Glu205,	Phe357, Tyr547, Tyr666, Lys554
Saxagliptin	Ser209, Ser630, Arg125, Tyr547	Tyr547	Glu206, Glu205, Tyr631, Tyr662, His126, His740, Arg358	Phe357, Tyr666
Alogliptin	Arg125, Arg358, Glu205, Glu206, Asn710	Ser209, Arg358	Tyr662	Phe357, Tyr 666

Hb: hydrogen bond; CHb: carbon–hydrogen bond; VdW: Van der Waals interaction; π–π: pi–pi bond.

**Table 9 ijms-20-05913-t009:** Physicochemical parameters of potential DPP-IVi hits identified from *Nauclea latifolia* extracts and their comparison with Lipinski rule details.

	2-*O*-p-Methylphenyl-1-thio-β-d-glucoside	3-Tosyl sedoheptulose	4-Benzyloxy-6-hydroxymethyl-tetrahydropyran-2,3,5-triol	Vitamin E	Lipinski Rule Details
MW (g/mol)	286.34	364.37	270.281	430.71	≤ 500
Hb donor	4	5	4	1	≤ 5
Hb acceptor	5	9	6	2	≤ 10
LogP	−0.113	−2.137	−0.997	8.84	≤ 5
TPSA	115.45	162.13	99.38	29.46	-
NRb	3	5	4	12	-
MR	70.78	79.25	64.95	139.27	-
# Atoms	37	44	37	81	-
# Lipinski Violations	-	-	-	1	

MW: molecular weight; Hb: hydrogen bond; LogP: octanol-water partition coefficient; TPSA: topological polar surface area; NRb: number of rotatable bonds; MR: molar refractivity.

**Table 10 ijms-20-05913-t010:** Predicted pharmacokinetic and toxicity properties of potential DPP-IVi hits identified from *N. latifolia* extracts.

	2-O-p-Methylphenyl-1-thio-β-D-glucoside	3-Tosyl sedoheptulose	4-Benzyloxy-6-hydroxymethyl-tetrahydropyran-2,3,5-triol	Vitamin E
**Absorption**				
Caco-2 permeability (cm/s)	− (−5.697)	− (−6.137)	− (−5.745)	+ (−4.969)
Blood Brain Barrier	++	−	++	+++
Human Intestinal Absorption	−−−	−−−	−−−	++
P-glycoprotein Inhibitor	−−−	−	−−−	−
P-glycoprotein Substrate	−−−	−−−	−−−	−−−
F (20% Bioavailability)	+	−	−	−
F (30% Bioavailability)	−	−	−	−
Renal Organic Cation Transporter	−	−	−	−
**Distribution**				
Subcellular localization	Mitochondria	Lysosome	Mitochondria	Mitochondria
Plasma Protein Binding (%)	55.94	49.26	43.47	84.65
Volume Distribution (L/kg)	−0.496	−1.017	−0.278	0.444
**Metabolism**				
P450 CYP1A2 inhibitor	−−−	−−−	−−−	−−−
P450 CYP1A2 Substrate	−−−	−−−	−−−	−
P450 CYP3A4 inhibitor	−−−	−−−	−−−	−
P450 CYP3A4 substrate	−	−	−−−	++
P450 CYP2C9 inhibitor	−−−	−−−	−−−	−−−
P450 CYP2C9 substrate	−	+	−	+
P450 CYP2C19 inhibitor	−	−	−−−	−−−
P450 CYP2C19 substrate	−	−	−	+
P450 CYP2D6 inhibitor	−−−	−−−	−	−−−
P450 CYP2D6 substrate	−	−	−	−
CYP Inhibitory Promiscuity	Low	Low	Low	Low
**Excretion**				
T_1/2_ (h)	0.98	1.21	0.93	1.95
Clearance Rate (mL/min/kg)	1.550	0.724	1.732	1.581
**Toxicity**				
Human Ether-a-go-go-Related Gene Blockers	−	−	−	−
AMES Mutagenicity	−−−	−−−	−−−	−−−
Skin sensitization	−	−	−−−	−
LD_50_ (mg/kg)	736.03	1146.95	1967.05	1161.96
FDA Maximum Recommended Daily Dose	++	++	++	+
Carcinogens	Non-carcinogen	Non-carcinogen	Non-carcinogen	Non-carcinogen
Acute Oral Toxicity	III	III	III	III

“−−−”, “−−”, “−”, “+”, “++” and “+++” signify the level of predicted pharmacokinetic and toxicity property.

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
