# Peer review of "Model Optimization and In Silico Analysis of Potential Dipeptidyl Peptidase IV Antagonists from GC-MS Identified Compounds in Nauclea latifolia Leaf Extracts"

_ijms, 2019, doi:10.3390/ijms20235913_

Round 1

Reviewer 1 Report

line 16In common sense “binding pockets” were directed to some concrete protein residues which were used in molecular docking from target protein just as DPP-IV. so the protein residue which binding pocket interacted to molecules should be assigned.  

line 22: Abbreviation should not be provided in its first presence.

line 81: Many components in aqueous fraction from plant are not volatile and not to be analyzed by GC-MS, and these non-volatile components will damage the column. Explained it.

line 97-101: Many compounds in table 1 an table 2 showed bioactivities, but evidences of bioactivities were not complemented in any parts of this work, or provided from reported articles. Explain it.

line 162: it is not DPP-IV biding pocket, it should be a binding pocket in a protein residue from DPP-IV. check it.

line 191 and line 199: explain these binding poses of acceptor and ligands. Are the acceptors same residue from DPP-IV, or different residues from DPP-IV?

line 212: MW(g/mol), correct it.

line 352:"a fused Optima-5MS capillary column of 30 mm length". a capillary column is only 30mm long? check it.

Author Response

Both chain A and B of DPP-IV are similar in terms of structure and amino acid residues. Thus, the authors preferred to state the chain which was used for docking in the body of the manuscript rather than the abstract.

Full meaning has been placed in first place abbreviation

The sample was derivatised to ensure polar non-volatile compounds were analysed by GC-MS and prevent damage to the column.

The identified bioactivities of the compounds were from Dr Duke’s: Phytochemical and Ethnobotanical Databases which has been added. These bioactivities are broad-based and the reason most of the pieces of evidence were not complemented is the specific activity (DPP-IV antagonist) that is being studied.

The identified druggable binding pocket was identified in the DPP-IV protein itself, not from an identifiable protein residue from DPP-IV

iGEMDOCK uses a different binding methodology from Autodock Vina, hence there are interactions in other smaller allosteric pockets different from the active site detected by iGEMDOCK.

The authors humbly differ as molecular weight is correctly written and values have a unit of gram per mole (g/mol)

The column has been corrected to 30 m

Reviewer 2 Report

the authors present results on the characterization on some metabolites from Nauclea latifolia leaf extracts and the effect on DPP-IV enzyme.   major negative points: the authors modelled the NP_001926.2 sequence on 1cwy.pdb used as template. the two proteins are IDENTICAL !!!, there is no need of homology modelling in such a case. In table 3, the authors admit that the sequence identity is 100 (%)…   minor points: - the authors should include some representative MS spectra of the identified compounds. - possibly the paper should include some functional binding studies, validating experimentally the predicted affinity constants between DPP-IV and the identified compounds.  

Author Response

Homology was important to build a high quality 3D DPP-IV structure. The authors noted a sequence identity of 100% for not only 1wcy but also 2bgr, 2jid and 3f8s. Taking other parameters such as sequence similarity, global model quality estimate, template resolution, quaternary structure quality estimate (QSQE), oligomeric state, local quality estimate and experimental comparison plot superiority over other templates into consideration the model 1wcy was the best in all parameters and thus the modelled protein from 1wcy was selected.

Chemical formula and molecular weight are some representative MS spectra of the identified compounds that have been included

Meduru et al (2016) has reported on a correlation between predicted affinities and experimental studies which has been included in the manuscript. However, To the best of our knowledge, there is no functional binding study validating experimentally the predicted affinity constants between DPP-IV and the identified compounds as this has also been recommended by the authors for further studies. 

Reviewer 3 Report

The abstract needs to be rewritten. Some of the long sentences need to be corrected and separated.

There should be an overview of the author's hypothesis related to the current study needs to be added in the introduction section.

It is confusing that authors modeled the DPP-IV sequence using 1WCY template. Also, they mentioned there are various PDB structures available in the PDB database. Why they have not selected the crystal?

The protein sequence retrieval and homology modeling results section just explained the technical details and not related to any of the results and discussion. This should be clearly explained.

The physicochemical properties of amino acids are nowhere related to basic science research. It is not related to the current context of the study. Also, these properties are always there for a given protein. Authors should focus on their research findings part instead of amino acid general terms.

Docking analysis is just giving numerical values of binding and not at all correlating with the findings. This should be critically checked and interpreted in the text.

Author Response

Abstract has been rewritten. Long sentences have been corrected and separated

The authors humbly differ in including hypothesis as a short statement of research problem and justification of study was included in the introduction.

Homology modelling was important to build a high quality 3D DPP-IV structure. The authors noted other structures asides 1wcy such as 2bgr, 2jid and 3f8s in the database. However, taking other parameters such as sequence similarity, global model quality estimate, template resolution, quaternary structure quality estimate (QSQE), oligomeric state, local quality estimate and experimental comparison plot superiority over other templates into consideration, the modelled DPP-IV protein from 1wcy was selected and considered best in accordance with these parameters.

The results and discussion of the protein sequence retrieval and homology modelling has been properly related to the results and discussion

The authors did not determine the physicochemical properties of amino acids rather, it was the physicochemical properties of the modelled protein.

Docking analysis was extended past numerical values to show the various interactions between the hit compounds and the various amino acid residues. It was based on these interactions that inference was made and corroborated by other studies that carried out similar work.

Round 2

Reviewer 2 Report

In the authors' response, the authors claim that "Homology was important to build a high quality 3D DPP-IV structure". There is no need to use homology modeling to improve an experimentally obtained structure (1wcy). Just as proof, I personally submitted 1wcy to the optimization procedure used by the authors (3Drefine), and the minimised structure has been analyzed using the ERRAT section in the Verify3D server:

As you can see, the ERRAT overall quality factor (96.671) is higher than the one obtained by the authors using homology modeling and then minimizing the obtained model (92.93 see fig.7)

In conclusion, the homology modeling section should be eliminated from the text. The authors should reshape the text accordingly.

The authors should include the MS spectrum, not MW values, for some of the chromatographic peaks identified (for instance peak 14 in fig.2 looks quite unresolved, so its MS spectrum should show in agreement a complex mix of several compounds). 

Author Response

The ERRAT quality of optimized 1wcy is not so far from the value (92.93%) obtained for the minimised modelled DPP-IV. Also note the Verify3D value (95.34%) of the optimized modelled DPP-IV was higher than 1wcy (94.47%).

These values (> 90%) are a pointer that the modelled protein not only has an overall good structural quality but also very good stereochemical quality as shown in table 5.

The authors will like to refer to the following underlisted publications that have lower errat and Verify3D values for their modelled proteins compared to their templates

http://dx.doi.org/10.1155/2013/520435 https://doi.org/10.1016/j.imu.2018.06.007 https://doi.org/10.1016/j.compbiolchem.2018.11.002

MS spectrum has been added as supplement.

Reviewer 3 Report

I agree with the revised version of the manuscript. Thanks!

Author Response

The reviewer agrees with the revised version of the manuscript. Thanks!

Round 3

Reviewer 2 Report

The authors try to convince the reviewer that "The ERRAT quality of optimized 1wcy (96.671%) is not so far from the value (92.93%) obtained for the minimised modelled DPP-IV. Also note the Verify3D value (95.34%) of the optimized modelled DPP-IV was higher than 1wcy (94.47%)."

So 96.671% is not far from 92.93%, while 95.34% is higher than 94.47%...

I tried to explain that you cannot use homology modeling to get a model of an experimentally-obtained template (1wcy). It is worthless.

The authors used the SwissModel server to obtain a more reliable DPP-IV model than the template.

Models are computed by the SWISS-MODEL server homology modelling pipeline (Waterhouse et al.) which relies on ProMod3, an in-house comparative modelling engine based on OpenStructure (Biasini et al.).After templates search, and sequence modelling, the small structural distortions, unfavourable interactions or clashes introduced during the modelling process are resolved by energy minimisation. ProMod3 uses the OpenMM library (Eastman et al.) to perform the computations and the CHARMM27 force field (Mackerell et al.) for parameterisation.

So the authors are using the SwissModel optimization routines to obtain a reliable DPP-IV model. The search for homologous templates is worthless, and the use os SwissModel as structure optimizer is possible, but again is not the purpose of homology modelling.

So I strongly suggest that the authors state that they used an homology modeling approach just to use the energy minimisation routines used by SwissModel.

The paper title should be changed accordingly. Instead of "Homology modeling..." they should use "Model optimization ...

Author Response

Authors have stated that they used an homology modeling approach just to use the energy minimisation routines used by SwissModel.

The paper title has been changed accordingly from "Homology modeling..." to "Model optimization ...